# Immunotherapy Moves to the Early-Stage Setting in Non-Small Cell Lung Cancer: Emerging Evidence and the Role of Biomarkers

**DOI:** 10.3390/cancers12113459

**Published:** 2020-11-20

**Authors:** Xabier Mielgo-Rubio, Virginia Calvo, Javier Luna, Jordi Remon, Margarita Martín, Pedro Berraondo, José Ramón Jarabo, Oliver Higuera, Esther Conde, Javier De Castro, Mariano Provencio, Florentino Hernando Trancho, Fernando López-Ríos, Felipe Couñago

**Affiliations:** 1Department of Medical Oncology, Hospital Universitario Fundación Alcorcón, Budapest 1 Alcorcón, 28922 Madrid, Spain; 2Department of Medical Oncology, Puerta de Hierro Hospital, Joaquín Rodrigo 1, Majadahonda, 28222 Madrid, Spain; virginia.calvo@salud.madrid.org (V.C.); mariano.provencio@salud.madrid.org (M.P.); 3Department of Radiation Oncology, Fundacion Jimenez Diaz, Oncohealth Institute, Avda. Reyes Católicos 2, 28040 Madrid, Spain; jluna@fjd.es; 4Department of Medical Oncology, Centro Integral Oncológico Clara Campal (HM-CIOCC), Hospital HM Delfos, HM Hospitales, 08023 Barcelona, Spain; jordi.remon@delfos.cat; 5Department of Radiation Oncology, Ramón y Cajal University Hospital, M-607, 100, 28034 Madrid, Spain; margarita.martin@salud.madrid.org; 6Division of Immunology and Immunotherapy, Cima Universidad de Navarra and Instituto de Investigacion Sanitaria de Navarra (IdISNA), 31008 Pamplona, Spain; pberraondol@unav.es; 7Department of Thoracic Surgery, Hospital Clínico San Carlos, Calle del Prof Martín Lagos, s/n, 28040 Madrid, Spain; joseramon.jarabo@salud.madrid.org (J.R.J.); florentino.hernando@salud.madrid.org (F.H.T.); 8Department of Medical Oncology, Hospital Universitario La Paz, Paseo de la Castellana 261, 28046 Madrid, Spain; oliver.higuera@salud.madrid.org (O.H.); javier.decastro@salud.madrid.org (J.D.C.); 9Pathology-Targeted Therapies Laboratory, HM Hospitales, 28015 Madrid, Spain; econde@hmhospitales.com (E.C.); flopezrios@hmhospitales.com (F.L.-R.); 10Department of Radiation Oncology, Hospital Universitario Quirónsalud Madrid, Pozuelo de Alarcón, 28223 Madrid, Spain; felipe.counago@quironsalud.es; 11Department of Radiation Oncology, Hospital La Luz, 28003 Madrid, Spain; 12Department of Radiation Oncology, Universidad Europea de Madrid, Villaviciosa de Odón, 28670 Madrid, Spain

**Keywords:** immunotherapy, early-stage, non-small cell lung cancer, biomarkers, PD-1, nivolumab, pembrolizumab, atezolizumab, durvalumab

## Abstract

**Simple Summary:**

In recent years there has been a trend towards an increase in the proportion of non-small cell lung cancer patients diagnosed with localized stage instead of advanced. However, 5-year survival rates continue to be low, even among patients diagnosed at early stages. In recent years major advances have been made in the treatment of advanced NSCLC, in large part due to the irruption of immunotherapy. PD-1 axis blocking-based immunotherapy is already a well-established standard of care treatment for patients with advances NSCLC, in frontline setting and in pretreated patients. Our greatest challenge now is to move the benefit of immunotherapy to patients with early-stage NSCLC so as to increase 5-year survival rate. The aim of this manuscript is to make a comprehensive review of available evidence, make a critical review of the results of published and ongoing studies, and analyze the role of biomarkers, main areas of controversy and future challenges.

**Abstract:**

Despite numerous advances in targeted therapy and immunotherapy in the last decade, lung cancer continues to present the highest mortality rate of all cancers. Targeted therapy based on specific genomic alterations, together with PD-1 and CTLA-4 axis blocking-based immunotherapy, have significantly improved survival in advanced non-small cell lung cancer (NSCLC) and both therapies are now well-established in this clinical setting. However, it is time for immunotherapy to be applied in patients with early-stage disease, which would be an important qualitative leap in the treatment of lung cancer patients with curative intent. Preliminary data from a multitude of studies are highly promising, but therapeutic decision-making should be guided by an understanding of the molecular features of the tumour and host. In the present review, we discuss the most recently published studies and ongoing clinical trials, controversies, future challenges and the role of biomarkers in the selection of best therapeutic options.

## 1. Introduction

Lung cancer is the most common type of cancer worldwide, with 2.1 million new cases annually, and also the leading cause of cancer-related mortality (1.8 million deaths in 2018) [1]. Non-small cell lung carcinoma (NSCLC) accounts for approximately 85% of lung tumours. NSCLC has a poor prognosis, posing a serious health risk even in patients with early stage disease, with a low 5-year survival rate [2]. Although most patients are diagnosed with advanced disease (48.7% in 2015 according to the SEER database), better diagnostic techniques and widespread screening may be the key to achieving an earlier diagnosis. In fact, there has been a clear trend in recent years towards an increase in the percentage of patients diagnosed with localized NSCLC, from 16.6% in 1988 to 23.6% in 2015 (SEER database) [3]. 

Major advances have been made in the treatment of NSCLC in recent years, leading to a significant improvement in survival outcomes [4]. Most of these treatment advances have occurred in advanced disease due to the discovery of a number of oncogenic mutations (unrelated to tobacco use) responsible for some lung tumours. The discovery of these molecular pathways has led to the development of targeted anti-cancer drug therapies, with excellent results in terms of antitumour efficacy. The first oncogenic mutation identified, in the year 2004, was the epidermal growth factor receptor (EGFR) mutation [5,6]. However, numerous other mutations have been discovered, including ALK, ROS1, BRAF, MET, RET, and NTRK, among others [7]. Indeed, the improved survival outcomes in patients with lung cancer observed through the year 2016 correspond closely with the timing of regulatory approval of targeted therapies. In the coming years, additional improvements in survival outcomes are expected due to the introduction of immunotherapy, which has been used in clinical practice to treat advanced NSCLC since 2015 with PD-1 and CTLA-4 axis blocking-based monoclonal antibodies (mAbs). Together, targeted therapies and immunotherapy represent a major paradigm shift in the treatment of NSCLC [8]. (Table 1).

Advanced NSCLC refers to those patients with metastatic NSCLC and treatment objectives focus on prolonging survival and improving quality of life of these patients. On the other hand, early-stage NSCLC comprises those tumours between stages I and III of the TNM classification system developed by the American Joint Committee on Cancer (AJCC) and treatment aim is curative. We need to add surgery and/or radiotherapy to achieve this goal so far in early-NCSLC.

At present, immunotherapy is approved only for the treatment of advanced NSCLC, with the notable exception of consolidation durvalumab, which has been approved to treat unresectable locally-advanced NSCLC after radical chemoradiation (CRT). In advanced setting, immune checkpoint inhibitors (ICI) based immunotherapy has demonstrated overall survival (OS) improvement both in palliative first line and second line setting. In unresectable stage III NSCLC, consolidation durvalumab improved disease-free survival (DFS) and OS after radical CRT.

Antitumour effect of ICI-based immunotherapy is based on enhancing the ability of the host’s immune system to recognize tumour cells as strange to trigger an antitumour immune response that ends up eliminating the tumour cells. The presence of tumour cells with neoantigens different from the normal origin cells are present from the beginning of tumour growth, both in early and advanced stages, so there is biological rational for immunotherapy to be also effective in early stages.

Numerous studies are already underway to assess the role of these treatments in early-stage NSCLC, with early results supporting this therapeutic approach in these patients, which is particularly relevant given that early treatment could have the greatest impact in terms of reducing mortality rates [9]. Presently, our greatest challenge is to make the demonstrated benefit of immunotherapy in advanced disease available to patients with localized or locoregional disease [10]. In the present article, we review the data supporting the implementation of immunotherapy in early-stage NSCLC. We discuss the results of published studies as well as clinical trials currently in progress and role of biomarkers. Finally, we critical review the main areas of controversy and future challenges.

## 2. Immune Checkpoint Inhibitors in Early NSCLC

In every step of the tumorigenesis process, tumours must overcome the body’s antitumour effector immune response. To avoid the effects of the immune system, tumours deploy a multitude of immune escape mechanisms. Recent clinical evidence shows the relevance of one of these mechanisms present in multiple cancer types, including NSCLC: the PD-1/PD-L1 pathway. Inflammatory cytokines such as interferon-gamma induce PD-L1 expression, both in tumour cells and in myeloid-derived cells infiltrating the tumour microenvironment. Interaction between PD-L1 and PD-1, expressed on activated T lymphocytes, limits the proliferation, activation, and effector mechanisms of tumour-specific T lymphocytes. The remarkable clinical efficacy of mAbs targeting PD-1 or PD-L1 has led to the approval of these agents as monotherapy or combination therapy in different stages of NSCLC. However, primary and secondary resistance is frequent, thus limiting the long-term clinical benefit of these treatment modalities [11]. However, combination therapies designed to synergize with anti-PD-1/PD-L1 mAbs can overcome these resistance mechanisms. The antitumour efficacy of mAbs targeting the PD-1 pathways relies on a pre-existing antitumour effector immune response. Therefore, therapeutic strategies that prime an antitumour immune response may synergize with anti-PD-1/PD-L1 mAbs. In this regard, CTLA-4 stands out as a critical immune checkpoint during the priming phase of the immune response. CTLA-4 prevents CD28 signalling required for efficient activation of effector T lymphocytes [12]. The combination of anti-PD-1 and anti-CTLA-4 promotes an immune response characterized by an increase of CD4+ICOS+T lymphocytes and a different CD8+ population than the exhausted CD8+PD-1+TIM-3+LAG3+T lymphocytes that dominate the response after anti-PD1 monotherapy [13].

Conventional cancer therapies can complement anti-PD-1/PD-L1 mAbs, thus providing early control of disease progression. However, in addition to early clinical control, surgery, chemotherapy, and radiotherapy also enhance the priming phase of the immune response, and can, therefore, synergize with anti-PD-1 mAbs.

Regarding surgery, preclinical research has demonstrated the superiority of neoadjuvant immunotherapy over adjuvant immunotherapy. Neoadjuvant immunotherapy is characterized by an increase in tumour-specific CD8+cells, suggesting a role in T cell priming or in T cell migration into the tumour, although further research is needed to decipher immune mechanisms implicated. Many more preclinical studies have evaluated the potential of chemotherapy and radiotherapy combined with immunostimulatory monoclonal antibodies. Several chemotherapy and radiotherapy regimens induce immunogenic cell death, characterized by tumour-associated antigen release in the context of danger signals that promote the activation of the cross-presenting dendritic cells, characterized by the expression of the Batf3 transcription factor. In addition, elimination of immunosuppressive immune populations, such as T regulatory cells and myeloid-derived suppressor cells, has also been frequently reported for a variety of chemotherapy and radiotherapy regimens. In the case of radiotherapy (RT), a robust abscopal effect has been observed in preclinical models using ablative hypofractionated radiation dose schedules, such as 8 Gy × 3 fractions. These RT regimens allow cytoplasmic DNA accumulation and subsequent activation of the cGAS-STING (stimulator of interferon genes) pathway, leading to high type I interferon production. In contrast, a single dose of 20 Gy induced the expression of the exonuclease Trex, which prevents the accumulation of cytoplasmic DNA and induction of immune response priming [14].

## 3. Immunotherapy in Early-Stage NSCLC: Combining it with Surgery and Radiotherapy

### 3.1. Resectable/Potentially-Resectable Early-Stage NSCLC

#### 3.1.1. Current and Emerging Evidence with ICIs

Surgery is the main treatment option in patients with resectable localized NSCLC, with 5-year OS rates ranging from 36% (stage IIIA, with N2 detected incidentally during surgery) to 92% (stage IA1) [15]. Numerous studies have been performed in an effort to improve these outcomes by adding adjuvant treatment, with the first of these studies conducted in the 1980s with contradictory findings. A meta-analysis carried out by the Non-Small Cell Lung Cancer Collaborative Group (NSCLC-CG), published in 1995, evaluated 14 studies (4,357 patients) comparing surgery alone to surgery followed by chemotherapy (CT) [16]. After publication of that meta-analysis, several randomised trials were initiated. The largest and most important study of adjuvant therapy, which changed the treatment of these patients, was the International Adjuvant Lung Cancer (IALT) study, published in 2004. That trial demonstrated an improvement in OS in patients who received adjuvant CT (cisplatin doublet) [17]. The results of the LACE (Lung Adjuvant Cisplatin Evaluation) meta-analysis, which evaluated five studies (ALPI, IALT, BLT, JBR.10, and ANITA) comprising 4584 patients, were published in 2008 [18], showing that patients who received adjuvant CT had a 5.4% improvement in OS at 5 years (median follow-up: 5.2 years) and an 11% reduction in the risk of death (HR 0.89, 95% CI 0.82–0.96, *p* = 0.0043).

Survival rates in patients surgically-treated for NSCLC remain poor, underscoring the need for novel therapeutic strategies. Multiple clinical trials are currently underway in early-stage NSCLC to assess the role of ICIs in both adjuvant and neoadjuvant settings. Initially, several vaccines were evaluated, with one study demonstrating that immunotherapy against the tumor-specific MAGE-A3 antigen in melanoma showed anti-tumour activity [19]. Based on these findings, it was proposed to evaluate this vaccine in lung cancer. Adjuvant MAGE-A3 immunotherapy was first evaluated in a randomised phase 2 trial in patients with completely-resected stage IB-II NSCLC [20]. In a subsequent phase 3 trial (MAGRIT) [21], 2227 MAGE-A3 positive patients with completely-resected stage IB, II, and IIIA NSCLC were randomised, in a 2:1 ratio, to receive the MAGE-A3 vaccine or placebo. However, no significant differences were observed in DFS rates (60.5 vs. 57.9 months, HR 1.02). Currently, five phase 3 clinical trials are underway to evaluate the role of immunotherapy in patients with completely-resected NSCLC. In those trials, more than 4500 patients with stage IB-IIIA NSCLC have been randomised (regardless of PD-L1 status) to receive one year of ICI or placebo (PEARLS, BR31, CANOPY-A) or one year of ICI vs. observation (ANVIL and IMpower-010) after standard CT, if indicated. The main outcome measure in all five trials is DFS (Table 2). 

Neoadjuvant and adjuvant therapy have a comparable impact on OS outcomes [22], although adjuvant therapy is supported by a larger body of evidence. Neoadjuvant therapy has numerous advantages, and it is an excellent clinical scenario to identify clinical and molecular markers. Several phase 3 trials have demonstrated that platinum-based induction CT improves OS in NSCLC, including patients with stage IIIA (N2) disease [23,24]. These results were confirmed in a subsequent meta-analysis [22].

Immunotherapy administered in combination with neoadjuvant therapy could potentially induce an antitumour immune response that persists beyond surgery, thus preventing recurrent disease. Indeed, various studies have demonstrated the feasibility and safety of ICI as a neoadjuvant therapy in NSCLC (Table 3).

In the trial carried out by Forde et al. (NCT02259621) [25], administration of nivolumab was not associated with a delay in surgery nor an increase in perioperative complications. In that trial, treatment-related adverse events (AE) of any grade were observed in 23% (5/22) of patients, with only one AE ≥ grade 3. In addition, major pathological response (MPR) was observed regardless of PD-L1 expression. The tumour mutational burden (TMB) was predictive of pathological response. At a median follow-up of 30 months, the median recurrence-free survival (RFS) had not been reached, with a 24-month RFS of 69% (95% CI: 51–93) [26]. In the LCMC3 trial (NCT02927301) [27] one patient developed a grade 5 AE (not treatment related) while 16 presented grade 3/4 AEs (three of which were treatment-related). Surgery was delayed in one patient due to grade 3 immune-related pneumonitis. The NEOSTAR study (NCT03158129) is a phase 2 trial involving patients with stage I-IIIA (N2 only) NSCLC. Patients received three doses of nivolumab 3 mg/kg IV every 2 weeks or nivolumab 3 mg/kg IV every 2 weeks for three cycles, plus ipilimumab 1 mg/kg on day 1 followed by surgery. AEs were observed in 4% of patients, including 2 cases of bronchopleural fistula and 8 air leaks [28].

Studies demonstrating that chemoimmunotherapy is superior to CT alone in patients with metastatic NSCLC have prompted interest in evaluating the role of chemoimmunotherapy as neoadjuvant therapy in early-stage NSCLC followed by surgery. The NADIM study (NCT03081689) is an open-label, phase 2, single-arm clinical trial to evaluate the safety and efficacy of neoadjuvant CT (paclitaxel 200 mg/m^2^ + carboplatin AUC 6 IV every 3 weeks) plus nivolumab (360 mg IV), 3 cycles, followed by surgery and one year of adjuvant nivolumab (240 mg IV every 2 weeks for 4 months and 480 mg IV every 4 weeks for 8 months) in 46 patients with resectable stage IIIA (N2 or T4) NSCLC [29]. The primary endpoint is progression-free survival (PFS) at 24 months. The latest results of that trial were presented at the 2019 World Conference on Lung Cancer (WCLC), with the highest pathological complete response (pCR) rate observed to date in this patient population. At a median follow-up of 17 months, the 18-month PFS and OS rates were 81% and 91%, respectively [30]. A new randomised phase 2 clinical trial is underway (currently recruiting) to compare the same treatment regimen (neoadjuvant CT plus nivolumab followed by surgery) plus adjuvant nivolumab for 6 months or standard CT alone (NADIM II; NCT03838159). Shu et al. (NCT02716038) conducted an open-label, multicenter, single-arm phase 2 trial to evaluate combined CT treatment (carboplatin AUC 5 IV every 3 weeks plus nab-paclitaxel 100 mg/m^2^ on days 1, 8, and 15) and atezolizumab 1200 mg IV every 3 weeks for 4 cycles. The primary endpoint was MPR. Thirty patients were included and 17 (57%) achieved a MPR [31]. The most common grade 3–4 AEs were neutropenia, elevated transaminases, and thrombopenia. Several other studies are currently evaluating the role of ICIs with or without CT in the neoadjuvant setting (Table 4).

#### 3.1.2. Role of SBRT and ICIs

Stereotactic body radiation therapy (SBRT) has become the standard treatment for inoperable early-stage NSCLC (ES-NSCLC) [35]. SBRT is defined by the American College of Radiology (ACR) and the American Society for Radiation Oncology (ASTRO) as the use of very high radiation doses (>6 Gy/fraction) delivered in few fractions (≤5) [36], an approach that has unique radiobiological characteristics capable of generating a strong tumour response. SBRT involves the delivery of highly conformal radiation to the tumour, with control of respiratory and tumour movement and daily image acquisition, which makes it a highly efficacious treatment [37,38].

Various trials have reported excellent results with SBRT in ES-NSCLC. The phase 2 RTOG 0236 trial in patients with inoperable ES-NSCLC yielded an impressive 5-year local tumour control rate of 93% [39,40], with only minimal pulmonary toxicity [41]. Regional and distant relapse rates can be as high as 30% in patients with ES-NSCLC [40,42,43]. We still need to define the patients who would be good candidates for systemic treatment plus SBRT due to a high risk of recurrence based on histological findings [44], pretreatment standardized uptake values (SUV) on 18F-FDG PET imaging [45], and the gene expression profile [46].

Most patients with ES-NSCLC who are candidates for SBRT (but not surgery) cannot safely receive CT. Moreover, the combination of adjuvant CT plus SBRT has not shown positive results in frail patients with ES-NSCLC who have multiple underlying pathologies [47]. The combination of immunotherapy—which is generally better tolerated than CT [48]—and SBRT has been evaluated primarily in patients with metastatic disease, with promising clinical results [49,50]. In addition to better tolerance, SBRT + immunotherapy offers important synergistic benefits, as SBRT can produce local and systemic antitumour effects mediated by the immune system, a phenomenon known as the abscopal effect [51], which is stronger when these two treatments are combined [49]. SBRT can reduce the tumour burden, thus permitting greater activation of T lymphocytes to destroy micrometastatic disease [52]. RT has shown other immunomodulatory effects that could also be synergistic when combined with immunotherapy, including the following: enhanced MHC class I expression [53], which allows for better recognition of the tumour cell by T lymphocytes; upregulation of FAS receptors in the tumour cells, leading to greater infiltration of these cells by T lymphocytes [54]; increased expression of NKG2D ligands, which allows for greater action of natural killer cells [55]; and other effects currently under investigation.

Despite the proven clinical benefit of SBRT plus immunotherapy in metastatic disease, there is still no demonstrated benefit for this combined treatment in ES-NSCLC, although several phase 1/2 clinical trials are currently investigating this approach. NRG Oncology is carrying out an interesting phase 3 trial of durvalumab vs. adjuvant placebo after SBRT in patients with unresected ES-NSCLC (PACIFIC-4) [56].

Table 5 summarizes the studies that are currently underway to investigate immunotherapy combined with SBRT in early-stage NSCLC. Some of these trials are evaluated the combination of SBRT, immunotherapy, and surgery. The NCT03217071 study proposes irradiating only 50% of the tumour, which will allow us to determine the local effect of SBRT plus immunotherapy as well as its impact on distant disease. Other trials are aiming to determine the optimal SBRT dose to combine with immunotherapy, the optimal time to deliver the two treatments, and the duration of immunotherapy. For example, the University of San Diego is evaluating immunotherapy plus SBRT (4 fractions of 12.5 Gy and 5 fractions of 10 Gy). In that trial, anti-PD-L1 therapy is administered 24 to 48 h before RT. In the trial being performed by the Tibor Rubin VA Medical Center and the Davis University of California, SBRT is administered with the third cycle. In the various trials, the duration of immunotherapy ranges from 3 and 24 months.

### 3.2. Unresectable Stage III NSCLC

One-third of NSCLC patients have stage III disease at diagnosis. In these patients, the standard of care (SoC) is concurrent platinum-based chemotherapy and radiation [57]. Unfortunately, OS remains poor, with a median OS ranging from 20 to 26 months [58,59] and 3- and 5-year OS of 30% and 15%, respectively [57,58]. Moreover, none of the novel strategies employed to date—such as adding induction or consolidation CT, the incorporation of EGFR inhibitors, or higher dose RT—have been shown to improve the OS versus SoC [60].

RT may increase the production and presentation of tumour antigens, which may enhance the antitumour immune responses elicited by ICIs [61]. Preclinical data support the rationale for combining both strategies [62], leading to the launch of various trials to assess this hypothesis. The phase 3 PACIFIC trial assessed the role of durvalumab (10 mg/kg Q2W) versus placebo as consolidation treatment for one year in 713 patients without progression after CRT. Durvalumab significantly achieved both co-primary endpoints, PFS (17.2 vs. 5.6 months, HR 0.55, 95% CI: 0.44–67, *p* < 0.0001) [63] and OS (47.5 vs. 29.1 months, HR 0.71, 95% CI: 0.57–0.88), with a 3-year OS of 55% vs. 44% and 4-year OS of 49.6% vs. 36.3%, respectively [64,65]. Durvalumab also improved the response rate (RR) (30% vs. 17.8%, *p* < 0.001) [63] and decreased the incidence of new brain metastases (6.3% vs. 11.8%, respectively) [66]. Safety was similar in the durvalumab and placebo arms (grade ≥ 3 AEs: 30.5% vs. 26.1%, including pneumonitis, 3.6% vs. 2.4%), as were treatment discontinuation rates (15.4% vs. 9.8%) [63,66]. Moreover, the benefit of durvalumab was achieved without a detrimental effect on patient-reported outcomes [67]. Although risk of pneumonitis in the PACIFIC trial was low and not associated with baseline respiratory disorders, prior RT dose, or prior cisplatin or carboplatin use [68], careful patient selection and active surveillance is required, as real-world studies indicate a grade 3 pneumonitis rate of 14.3% [69].

Enrolment in the PACIFIC trial was not restricted to any specific PD-L1 expression threshold level, and PD-L1 status was not mandatory for inclusion. A prespecified exploratory analysis assessed the benefit of durvalumab according to PD-L1 expression ≥ 25% (by SP263 IHC assay). Of the 63% of patients assessable for PD-L1 expression, 35% and 67% had PD-L1 ≥ 25% or PD-L1 ≥ 1%, respectively. In patients with PD-L1 ≥ 25%, durvalumab improved PFS (HR 0.41; 95%CI: 0.26–0.65) and OS (HR: 0.50, 95%CI: 0.30–0.83), whereas in those with PD-L1 < 25%, it improved PFS (HR 0.59, 95% CI: 0.43–0.82) but not OS (HR: 0.89, 95% CI: 0.63–1.25) [70]. The European Medicines Agency (EMA) requested an additional exploratory post-hoc analysis using a 1% cut-off for PD-L1 expression. Although durvalumab improved PFS and OS in tumours with PD-L1 ≥ 1%, in the 148 patients with PD-L1 < 1%, durvalumab neither improved PFS (HR 0.73; 95%CI: 0.48–1.11) nor OS (HR: 1.14, 95%CI: 0.71–1.84) [69]. Based on these data, the Food and Drug Administration (FDA) approved durvalumab as a new SoC regardless PD-L1 expression in February 2018, whereas the EMA approval in September 2018 was limited to tumours with PD-L1 ≥ 1%. The efficacy of durvalumab is currently being evaluated in a real-world setting in the PACIFIC-R trial (NCT03798535) [70]. Similarly, the ongoing phase 3 PACIFIC5 trial (NCT03706690) is evaluating a flat dose of durvalumab (1500 mg Q4W) compared to placebo after concurrent or sequential CRT. PD-L1 status by SP263 is mandatory in this trial. The phase 2 PACIFIC6 trial (NCT03693300) is assessing durvalumab (1500 mg Q4W) after sequential treatment. A planned interim analysis from the BTCRC-LUN 16-081 phase 2 trial comparing consolidative treatment after CRT with nivolumab plus ipilimumab versus nivolumab resulted in a higher percentage of grade 3 AEs (44% vs. 32%, including pneumonitis 16% vs. 4%), which resulted in a higher rate of treatment discontinuation (40% vs. 16%) [71].

The combination of pembrolizumab and CRT was evaluated in the phase 2 LUN 14-179 [72] and KEYNOTE-799 trials [73], atezolizumab in the DETERRED trial [74], and nivolumab in the NICOLAS trial [75,76], all with promising results (Table 6). Finally, the ongoing phase 3 PACIFIC2 trial (NCT03519971) is assessing durvalumab administered concurrently with definitive CRT, but the control arm is only CRT alone, which is less than ideal as the future challenge is to assess the best treatment approach, either concurrent ICI versus consolidation, and to assess the best consolidation approach (ICI vs. ICI plus ICI). The phase 3 Checkmate 73L (NCT04026412) trial is evaluating all of these treatment approaches. Another important question is the optimal treatment duration for consolidation therapy, especially as only 43% of patients enrolled in PACIFIC trial were able to complete the planned one-year of therapy [68]. Finally the role of predictive biomarkers, such as PD-L1 expression, and prospective validation of minimal residual disease assessed by dynamic circulating tumour DNA (ctDNA) may help to personalise consolidation ICI strategy after CRT [77].

## 4. Future Challenges for ICI in Early-Stage Disease

### 4.1. Optimal Treatment Duration

The optimal duration of neoadjuvant or adjuvant treatment with ICIs is unknown. At present, treatment duration is based on data from clinical trials that have evaluated neoadjuvant and adjuvant therapy in NSCLC. Treatment duration is an important consideration due to its potential impact on patient quality of life and with respect to the cost. Currently, there is no evidence of any correlation between longer treatment duration and increased survival in advanced NSCLC [78,79,80,81]. Indeed, exploratory analyses have found long-term NSCLC survivors even among patients who did not complete all ICI cycles, although the available data are limited [79,82].

In terms of neoadjuvant therapy, the trials that have evaluated platinum-based induction CT combined with third-generation CT agents have generally administered three cycles of neoadjuvant CT, with one study using four cycles [83]. For this reason, three induction cycles have been traditionally administered in clinical practice. Similarly, most studies that include ICIs in the neoadjuvant therapy regimen also administer three cycles, although several have used 2 or 4 cycles [25,28,29,31,84]. Consequently, the number of cycles administered in clinical practice generally corresponds to the cycles used in the trial on which the selected treatment regimen is based. Several of the studies that have evaluated neoadjuvant immunotherapy [29,84] (in monotherapy or in combination with CT), as well as the ongoing phase 2 and 3 trials, generally administer adjuvant ICIs for one year after surgery [32,33,34,85,86]. However, there is no concrete evidence to support this strategy, which is why it should be evaluated prospectively in randomised trials. In addition, the duration of adjuvant ICI presents other challenges in terms of treatment compliance and costs. Similarly, the optimal duration of adjuvant ICI treatment in patients who have not undergone prior induction therapy is not known. Most studies that have evaluated adjuvant CT have administered four cycles; however, the protocols of studies currently underway to assess adjuvant ICI as monotherapy without prior induction generally stipulate one year of ICI administration after standard adjuvant CT, with the exception of the BR.31/LINC trial, in which the duration is 6 months. Another unresolved question is whether it would be possible, in certain cases, to shorten the duration of adjuvant ICI in patients who have received neoadjuvant ICI therapy, or whether adjuvant ICI could be obviated in patients who achieve a pCR. New biomarkers, such as ctDNA, could potentially facilitate treatment decision–making in this clinical scenario [77].

### 4.2. Optimal Timing of Surgery

No evidence is available about the optimal timing of surgery after neoadjuvant treatment. The interval between the first neoadjuvant dose and surgery has varied in the different clinical trials. Thus, surgery was performed two weeks after the second cycle in the first trial of nivolumab, 3–4 weeks after the 21st day of the third cycle in the NADIM trial, and on day 29 after the 2nd cycle of pembrolizumab in the NEONUM trial. However, experimental analyses suggest that the efficacy of neoadjuvant immunotherapy in terms of survival may be dependent on an optimal duration between the first dose and resection [87]. The only study correlating the timing between neoadjuvant therapy and surgery is the study conducted by Gao et al. [88]. Those authors found that patients with resectable N2-IIIA who underwent surgery within 6 weeks after completing neoadjuvant CRT had significantly better OS than those who underwent surgery after six weeks. Traditionally, the optimal timing of surgery is between 4 and 6 weeks after completion of neoadjuvant therapy, based on histological changes secondary to radiation. However, this should not be extrapolated to new therapies without further, specific clinical research.

### 4.3. Surgical Challenges after Neoadjuvant Immunotherapy: New Patterns of Response

One difficulty that surgeons may face in patients who receive neoadjuvant ICI therapy prior to surgery is the response to immunotherapy, such as the contradictory response between the primary tumour and the hilar and mediastinal lymph nodes (probably due to genomic and immunological heterogeneity), in which an initial “tumour flare”, caused by immune cell infiltration, is observed. In these cases, it can be difficult to distinguish between pseudo-progression and real tumour progression. If this response is not interpreted correctly, surgery might be erroneously ruled out [89], a phenomenon that has been observed in up to 11% of patients with NSCLC who present nodal immune flare [90]. Although rare, hyperprogressive response patterns have been described in advanced disease [91]. This pattern could theoretically also occur in localized disease, although no cases have been reported to date. Consequently, the use of new radiological techniques, such as multiparametric magnetic resonance imaging [92] and/or positron-emission computed tomography (PET-CT), is important to better assess T-cell response [93] to differentiate between tumour response and progression in these clinical scenarios. Finally, evaluation of ctDNA levels [94] to assess tumour dynamics may also play a role in the future.

### 4.4. Challenges for Surgery with Neoadjuvant Immunotherapy: Surgical Difficulties

Most trials to date have focused on the complete resection rate, even though they agree that surgical morbidity and mortality do not differ from series without neoadjuvant therapies [29]. It is well-established among thoracic surgeons that surgical resection is technically more demanding after induction therapy, although it is difficult to quantify the degree of difficulty. Induction therapies induce tumour necrosis and the formation of scar tissue. The most challenging steps in the surgical procedure involve exposing the vascular structures to be sectioned and dissection of the hilar and mediastinal lymph nodes. The resection approach (i.e., minimally invasive vs. open) is a suboptimal way of evaluating the technical difficulty [95]. Changes in pulmonary structures after CT have been histologically documented [96]. Moreover, interstitial damage leading to a worsening in pulmonary tests directly related to higher postoperative complications has also been demonstrated [97,98]. In this regard, if we could predict the effects of new drugs, we could exclude patients with limited pulmonary function. Finally, it is essential to underscore the importance of using the term “complete resection” properly [99]. Complete resection requires the following: (i) free resection margins confirmed microscopically; (ii) systematic nodal dissection or lobe-specific systematic nodal dissection; (iii) absence of extracapsular nodal extension of the tumour; and (iv) the highest mediastinal node removed must be negative. If these four criteria cannot be met, then the resection must be considered uncertain. Complete resection defined in this way should be an inclusion criterion in clinical trials performed to evaluate surgical patients. For this reason, the involvement of thoracic surgeons in the design and development of these trials is mandatory.

### 4.5. Role of Biomarkers in Resectable NSCLC

Biomarker studies in early-stage tumours are approximately similar to those in advanced tumours. In advances setting most developed biomarkers are PD-L1 expression and TMB, and are the only ones that we use in daily clinical practice, but there are several biomarkers that have been or that are being studied. Neoadjuvant trials are an ideal setting for exploring predictive biomarkers and same markers as in advanced disease are being explored in resectable NSCLC, that include four major categories: tumour cell-associated biomarkers as PD-L1 expression and TMB, tumor microenvironment-related biomarkers, liquid biopsy-related biomarkers and host-related markers. We need to take into account that biomarkers in early-stage NSCLC have only been explored preliminarly and that we cannot confirm their value so far and even compare to their role in advanced disease. PD-L1 expression and TMB have not shown a consistent association with response to neoadjuvant immunotherapy. In the study by Forde and colleagues [25], tumours demonstrating a MPR to nivolumab were infiltrated with large numbers of lymphocytes and macrophages, and these changes were seen in both PD-L1-positive and negative tumours. As expected, tumours with a MPR had a higher TMB and a systematic increase in the number of T-cell clones in the tumour and peripheral blood. Interestingly, there were no alterations in immune-related genes (including CD274, PDCD1, CTLA4, B2M, and HLA) in patients with or without a MPR. In a phase 3 trial conducted by Shu and colleagues, PD-L1 expression did not appear to be predictive of a treatment benefit, and patients with STK11 tumour mutations did not have significant radiographic or pathological responses [31].

Both the NEOSTAR and LCMC3 trials found that immunotherapy showed activity (measured by MPR) against early-stage NSCLC. PD-L1 was positively correlated with MPR in NEOSTAR, but neither PD-L1 nor TMB correlated with MPR in LCMC3 [28,31]. Radiographic response was positively correlated with MPR in both studies.

T-cell expansion and ctDNA are emerging biomarkers that may prove useful in the future. In the CheckMate 159 trial, T-cell receptor (TCR) repertoire was significantly expanded in patients who achieved MPR and ctDNA clearance prior to surgery was detected in all patients who achieved a reduction ≥ 30% [100]. Furthermore, peripheral expansion of tumour-specific T-cells and long-term persistence were associated with longer DFS. In the NEOSTAR trial, a higher pretreatment TCR clonality in the blood was associated with a lower percentage of residual viable tumour at surgery in both treatment arms [101]. In the LCMC3 trial, the biomarker analysis based on paired peripheral blood samples showed significant increases from baseline in CD8+ T cells, mature NK cells, late-activated CD16+/CD56+ NK cells, CD16+ NK cells, and Th1 response-related dendritic cells. Those who did not achieve MPR showed significant increases in late-activated NK cells, a monocytic myeloid cell subpopulation, and a Th2- and Th17-response–related dendritic cell population. In the NADIM trial, a greater decrease in the platelets-to-lymphocytes ratio (PLR) post-treatment was associated with pCR (≥10% RVT). Moreover, higher pretreatment expression of PD-1 in CD4 T-cells and reduced activation on CD4 T and NK cells post-treatment are associated with pCR [102].

### 4.6. The Role of SBRT in the ICI Strategy

In early stage, non-operable NSCLC without nodal involvement, SBRT is the RT modality of choice. However, although SBRT achieves a local control rate of approximately 90%, lymph node and distant relapse rates range from 25% to 35% [39,40]. For this reason, proposals have been made to intensity treatment by offering systemic therapy in patients at high risk of nodal involvement or distant spread. Given the highly immunogenic nature of SBRT, together with the results achieved by combining SBRT and immunotherapy in metastatic patients and the better tolerance of immunotherapy compared to conventional CT, it would seem appropriate to offer the potential benefits of this combined therapy to patients with early stage but high risk disease: patients with micropapillary or solid histological subtypes, with a predominant mucinous component, vascular invasion, high SUV on PET-CT, and large peripheral or central cT2 tumours [44,45].

Although the tumour microenvironment is strongly immunosuppressive, administration of SBRT can alter this microenvironment, making it proinflammatory. Several studies have demonstrated that the antitumour effects of radiotherapy are at least partially based on activation of immunity [103], which produces a local anti-tumour effect, a bystander effect, and a distant effect (the abscopal effect). However, irradiation can also have an immunosuppressive effect; nodal irradiation, for example, could prevent the activation and accumulation of cytotoxic T lymphocytes and the adaptive immune response. In addition, high dose radiation could inhibit type I interferon, which would further support the combination of ICI with SBRT in tumours without nodal involvement, thereby avoiding nodal irradiation.

### 4.7. How Can We Improve the Results of Combined Immunotherapy/RT: Dose and Fractionation

At present, there are numerous unknowns, including the optimal dose and fractionation schedule required to achieve the immunogenic effect, the optimal manner of combining RT and immunotherapy, and how to best measure response. Golden et al. showed that immunogenic cell death depends on the dose per fraction [104,105]. Preclinical studies indicate that cell death is more likely at doses of 8–10 Gy per fraction [106], while doses greater than 15 Gy stimulate an increase in regulatory T lymphocytes (which inhibit the immune response) [107], and there is no effective immune activation at dose fractions less than 5 Gy. Thus, the preclinical data seem to indicate that there may be a dose threshold above which immunosuppression would prevail and below which there may be no significant immune system activation. The influence of the dose size on the emergence or not of an immune response could be explained by its effect on the STING pathway, which activates type I interferon. This pathway is a key component in the switch from the innate to adaptive immune response, since it allows for the recruitment of type 1 DCs. It is activated by the presence of DNA damaged by irradiation, in the cytosol. Vapouille-Box et al. found that TREX1, a DNA exonuclease, acts at high doses per fraction and degrades this cytosolic DNA, eliminating the stimulus for type I interferon activation [108,109], which would explain the absence of the abscopal effect at dose fractions above 15 Gy.

The duration of the immune response could also depend on the dose per fraction. At doses of 10 Gy, markers of immune activation are evident at 72 h, while PD-L1 expression is reduced 6 days after administration of SBRT [110]. Hettich and colleagues found that 2 fractions of 12 Gy each induced a transient increase in CD8+ cytotoxic T lymphocytes 5–8 days after irradiation, while immunosuppressive regulatory T cells were dominant on days 10 to 16 [111].

### 4.8. Is There Any Place for Surgery in Unresectable Stage III Disease at Present?

Until now, only curative-intent surgery had a role in NSCLC. However, paradigms of extended and unresectable disease have changed with the introduction of targeted therapies and immunotherapy in lung cancer [112]. The way these treatments sometimes achieve control of disease has made surgery becoming a complementary tool amenable to be considered in an increasing number of patients [113]. New questions that have emerged are the need to define which patients will benefit from surgery and the optimal time to perform the resection. At present, no data is yet available to answer these questions. The study of this patient cohort has evident limitations, including the following: heterogeneity in the factors that make the disease unresectable; local invasion criteria that are highly dependent on imaging data that is often imprecise; the application of multiple different therapies (CT, targeted therapies, immunotherapy, etc.) and multiple courses of treatment before resection. As a result, prospective trials will be difficult to design and retrospective data will need to be carefully assessed. Fortunately, the available data suggest that, even though the rate of pneumonitis secondary to long-term treatment is significant, overall postoperative complication rates (morbidity and mortality) are comparable to those observed in studies that have evaluated resection after neoadjuvant treatment regimens, and thus acceptable when compared to global surgical cohorts [114]. The limited evidence suggests that patients RT could cause specific histological changes and thus this subgroup of patients should be analysed separately. In terms of the type of resection, pneumonectomy should be avoided until we have greater experience. To obtain the maximum benefit from the multidisciplinary approach, the involvement of the thoracic surgeon throughout the whole disease process is essential, even if some patients will ultimately not undergo surgery.

### 4.9. Role of Biomarkers for ICI in Unresectable Localised NSCLC

Although the PACIFIC trial was not designed to evaluate durvalumab based on archival tumour PD-L1 expression, the results of exploratory analyses support a treatment benefit for durvalumab versus placebo irrespective of archival pre-specified tumour PD-L1 expression status. In that trial, the only patients who did not benefit in terms of OS from durvalumab were those with PD-L1 expression levels < 1%. However, this finding was based on an unplanned post hoc analysis with a PD-L1 cut-off level that differed from the original (25% vs. 1%) [69]. In the phase 2 DETERRED trial of atezolizumab with concurrent CRT, PD-L1 status was not associated with recurrence [74]. Furthermore, two patients developed a recurrence before the start of consolidation therapy: one had a KRAS/STK11 co-mutation and the other had an ALK rearrangement, a finding that suggests that molecular analysis in unresectable NSCLC would be of value to identify the patients expected to benefit or not from CRT/ICI combinations.

Moding and colleagues conducted a retrospective study to determine whether ctDNA, determine through a personalized profiling by deep sequencing (CAPP-Seq), could help to identify patients with NSCLC who might benefit from consolidation therapy with ICI after chemoradiation and also be used to monitor treatment response [77]. Those authors found patients with ctDNA detected after chemoradiation who then received consolidation ICIs had better PFS outcomes than patients with ctDNA (also detected post-chemoradiation) who did not receive consolidation immunotherapy. In addition, the data from that study suggest that the patterns of ctDNA levels may predict which patients are more likely to benefit from consolidation ICI: patients whose ctDNA levels begin to rise early in the consolidation ICI treatment had worse outcomes. In patients whose ctDNA levels continued to increase during the course of treatment developed progressive disease within 4.5 months of starting consolidation ICI, suggesting resistance to immunotherapy. Conversely, patients with decreasing ctDNA during consolidation ICI had good outcomes.

## 5. Conclusions

Immunotherapy and targeted therapy have revolutionized the treatment landscape in advanced NSCLC. For this reason, the role of these therapies in localised disease is current being studied, with promising results to date. However, in these early stages, administration of immunotherapy is more complex as their purpose is different, we look for the cure of the patient, so objectives are different. In this regard, surrogate markers of OS are needed to obtain more conclusive results earlier in the treatment process. In addition, we need to find the best way to combine it with radical RT and surgery, which is not an easy task, in part because there are still many unresolved questions in this area. In the adjuvant studies that are currently underway, the most common primary endpoint is DFS, rather than OS. Importantly, we lack predictive biomarkers and the optimal duration of adjuvant treatment remains unclear. We are currently awaiting the results of several trials evaluating the role of PD-1 axis blocking-based immunotherapy as an adjuvant therapy, although vaccine-based strategy failed to demonstrate survival benefit. In the neoadjuvant setting with immunotherapy, the combination of CT and immunotherapy appears to be more promising than immunotherapy alone, significantly increasing pCR rates. The studies conducted to date leave numerous unresolved questions, including the lack of predictive biomarkers and that we still do not know how to optimally assess radiological response or the optimal duration. However, we fully expect that ongoing trials will demonstrate a benefit for immunotherapy in early-stage disease as well. In short, it seems clear that immunotherapy (at least in patients without driver mutations) will inevitably form part of the treatment arsenal for early NSCLC in the near future based on the promising results of the studies published thus far and on the numerous trials currently in progress.

## Figures and Tables

**Table 1 cancers-12-03459-t001:** Characteristics of Immune Checkpoints Inhibitors discussed in the manuscript.

Name	Antibody Type	Mechanism of Action	Company
Nivolumab	Human IgG4	PD-1 inhibitor	Bristol-Myers Squibb
Pembrolizumab	Humanized IgG4	PD-1 inhibitor	MSD
Atezolizumab	Humanized IgG1k	PD-L1 inhibitor	Roche/Genentech
Durvalumab	Human IgG1k	PD-L1 inhibitor	Medimmune/Astra Zeneca
Ipilimumab	Human IgG1	CTLA-4 inhibitor	Bristol-Myers Squibb

**Table 2 cancers-12-03459-t002:** Ongoing phase 3 clinical trials with adjuvant ICIs.

Name	Trial Registration Number	Phase	Stage	N	Study Arm	Control Arm	Primary Objective	Trial Completion Date
**PEARLS/KEYNOTE-091**	NCT02504372	3	IB (≥4 cm)-IIIA	1080	Pembrolizumab 200 mg IV every 3 weeks for one year	Placebo, one year	DFS	2024
**BR31/LINC**	NCT02273375	3	IB (≥4 cm)-IIIA	1360	Durvalumab 10 mg/kg IV every 2 weeks for 6 months 20 mg/kg IV every 4 weeks for 6 months	Placebo, one year	DFS PD-L1+ DFS global	2024
**ANVIL**	NCT02595944	3	IB (≥4 cm)-IIIA	903	Nivolumab 240 mg IV every 2 weeks for 1 year	Observation	DFS OS	2024
**IMpower 010**	NCT02486718	3	IB (≥4 cm)-IIIA	1280	Atezolizumab 1200 mg IV every 3 weeks for one year	Observation	DFS II-IIIA DFS II-IIIA PD-L1+ DFS ITT	2027
**CANOPY-A**	NCT03447769		II-IIIA, IIIB (T > 5 cm and N2)	1500	Canakinumab 200 mg sc every 3 weeks for year	Placebo, one year	DFS	2027

ICI: immune checkpoint inhibitor; IV: intravenous; SC, subcutaneous; DFS: disease-free survival; OS: overall survival; ITT: intention-to-treat.

**Table 3 cancers-12-03459-t003:** Clinical trials of neoadjuvant ICI with or without chemotherapy.

Name	Registration Number	Phase	Stage	N	Resected	Treatment	Primary Objective	MPR (%)	Pcr (%)	Surgery (%)	Trial Completion Date
**SKCCC-JHU [25]**	NCT02259621	2	IB-IIIA	22	21	Nivolumab 3 mg/kg IV every 2 weeks, 2 cycles	Safety Feasibility	45	15	95	2023
**LCMC3 [27]**	NCT02927301	2	IB-IIIB (T3N2)	180	101	Atezolizumab 1200 mg IV every 3 weeks, 2 cycles	MPR	19	5	89	2020
**NEOSTAR [28]**	NCT03158129	2	I-IIIA (N2 only)	88	N: 23 NI: 21	N: 3 mg/kg IV every 2 weeks, 3 cycles NI: Nivolumab 3 mg/kg IV every 2 weeks, 3 cycles and Ipilimumab 1 mg/kg on day 1	MPR	N: 17 NI: 33	N: 9 NI: 29	N: 96 NI: 81	2021
**NADIM [29]**	NCT03081689	2	IIIA (N2 or T4)	46	41	CT+Nivolumab 360 mg IV every 3 weeks, 3 cycles—Postoperative nivolumab for one year	PFS at 24 months	83	59	89	2021
**Columbia University [31]**	NCT02716038	2	IB-IIIA	30	11	CT+Atezolizumab 1200 mg IV every 3 weeks, 4 cycles	MPR	57	33	87	2020

ICI, immune checkpoint inhibitors; N: Nivolumab; I: Ipilimumab; CT: chemotherapy; IV: intravenous; MPR: major pathological response; pCR: pathological complete response; PFS: progression-free survival.

**Table 4 cancers-12-03459-t004:** Clinical trials ongoing with neo/adjuvant ICI with or without chemotherapy.

Trial Name	Registration Number	Phase	Stage	N	Study Arm	Control Arm	Primary Objective	Estimated Completion Date
**KEYNOTE-671 [32]**	NCT03425643	3	IIB-IIIA	786	CT (CG or CP) + pembrolizumab 200 mg IV every 3 weeks, 4 cycles—pembrolizumab 200 mg IV every 3 weeks postoperatively	CT + placebo - and postoperative placebo	DFS, OS	2024
**CheckMate 816 [33]**	NCT02998528	3	IB-IIIA	350	CT + nivolumab 360 mg IV every 3 weeks, 3 cycles	CT, 3 cycles	DFS, pCR	2020
**IMpower 030 [34]**	NCT03456063	3	II-IIIA-IIIB (T3N2)	374	CT + atezolizumab 1200mg IV every 3 weeks, 4 cycles - Atezolizumab 1200 mg IV every 3 weeks postoperatively	CT + placebo - and postoperative placebo	MPR	2024
**Checkmate 77T**	NCT04025879	3	II-IIIB	452	CT + nivolumab 360 mg IV every 3 weeks, 4 cycles - nivolumab 480 mg IV every 4 weeks for one year postoperatively	CT + placebo - and postoperative placebo	DFS	2023
**AEGEAN**	NCT03800134	3	IIA-IIIA-IIIB (N2)	300	CT + Durvalumab 1500 mg IV every 3 weeks, 4 cycles -Durvalumab 1500 mg IV every 4 weeks, 12 cycles	CT + placebo - and postoperative placebo	MPR	2024
**SAKK 16/14**	NCT02572843	2	IIIA (N2)	68	CT, 2 cycles—Durvalumab 750 mg, 2 cycles—durvalumab 750 mg for 1 year		DFS	2021
**NADIM 2**	NCT03838159	2	IIIA-IIIB	90	CT + nivolumab 360 mg IV every 3 weeks, 3 cycles—nivolumab 480 mg IV every 4 weeks for 6 months postoperatively	CT	pCR	2022

ICI, immune checkpoint inhibitors; CT: chemotherapy; CG: cisplatin-gemcitabine; CP: cisplatin-pemetrexed; IV: intravenous; MPR: major pathological response; pCR: pathological complete response; DFS: disease-free survival OS: overall survival.

**Table 5 cancers-12-03459-t005:** Current clinical trials evaluating the combination of immunotherapy and SBRT.

Study Name	Study Type	Type of Patients	Treatment	Primary Objective	Secondary Objectives	Current Status
**STILE** **NCT03383302**	Phase 2, single arm, multicentric trial (Sponsor: Royal Marsden NHS Foundation Trust)	Stages I-II NSCLC	- SBRT (54 Gy in 3 fr of 18 Gy or 55 Gy in 5 fr of 11Gy) - Sequential nivolumab, 1 year	-Evaluation of lung toxicity	- Other toxicities -Local relapse rates, OS, DFS	Recruiting
**NCT03110978**	Phase 2, single arm trial (Sponsor: M.D. Anderson Cancer Center)	Stage I-IIA NSCLC	-SBRT (50 Gy in 4 fr or 70 Gy in 10 fr) -Nivolumab 12 weeks, started with 1st fraction of SBRT	-DFS	-OS -Adverse events -Analysis of immunological markers	Recruiting
**NCT03446911**	Randomised clinical trial (Sponsor: VU University Medical Center)	Stage I NSCLC	-ARM 1: SBRT with 2 cycles of pembrolizumab started on the 1st day of RT followed by lobectomy -ARM 2: SBRT without pembrolizumab followed by lobectomy	-Incidence and severity of adverse effects	-Expression of PD-1, PDL-1, CD4, among others	Unknown
**NCT02444741**	Randomised phase ½ clinical trial (Sponsor: M.D. Anderson Cancer Center)	NSCLC: early and advanced stages	Distinct groups included with varying combinations between pembrolizumab, SBRT or hypofractionated RT Pembrolizumab is started before SBRT (4 fr) or hypofractionated RT (15 fr). It is administered every 21 days until reach a maximum of 16 cycles	-Response rate and determination of radiological response -Toxicity Maximum tolerate dose of pembrolizumab	-DFS -OS	Recruiting
**SWOG S1914** **NCT04214262**	Phase 3 clinical trial (Sponsor: National Cancer Institute (NCI)	Stages I-IIA NSCLC	-ARM 1: Atezolizumab 8 cycles every 21 days. SBRT (3–5 fr) with cycle 3 of atezolizumab -ARM 2: SBRT (3–5 fr) at 21 days post- randomisation without atezolizumab	- OS	-SLP -Adverse effects	Recruiting
**PACIFIC-4/RTOG-3515** **NCT03833154**	Phase 3 multicentre, double-blind clinical trial (Sponsor: Astra Zeneca)	NSCLC stages I-II with negative nodes	-ARM 1: Durvalumab 1500 mg every 4 weeks up to 24 months of treatment or progression. SBRT (from 3–8 fr) ARM 2: Placebo an SBRT (from 3–8 fr)	-DFS	-OS -Lung cancer-specific mortality -Others	Recruiting
**ASTEROID** **NCT03446547**	Phase 2 multicentre, randomised clinical trial (Sponsor: Vastra Gotaland Region)	NSCLCT1-2N0M0	-Arm 1: SBRT (3–4 fr) -Arm 2: SBRT (3–4 fr) followed by durvalumab 1500 mg every 4 weeks 12 months	-TTP	-OS -Control local	Recruiting

Fr: fractions; OS: Overall survival; DFS: disease-free survival; PFS SLP: progression-free survival; TTP: Time to progression; RT, external beam radiotherapy.

**Table 6 cancers-12-03459-t006:** Summary of the efficacy of immune checkpoint inhibitors in stage III NSCLC.

Trial	Schedule	N	PFS	OS
**PACIFIC [63,65],**	CRT Durvalumab	713	17.2 m	3-y OS: 55% 4-y OS: 49.6% mOS: 47.5 m
**LUN 14-179 [72]**	CRT + P P	92	18.7 m.	3-y OS: 49% mOS: 36 m
**KEYNOTE 799 [73]**	CT CRT + P P	165	6-m PFS: 80%	
**NICOLAS [75,76]**	CRT+N N	79	12.4 m	1-y OS: 79%
**DETERRED [74]**	CRT CT + A A A + CRTCT + A A	10 30	18.6 m 13.2 m	22.8 m NR

N = number of patients; PFS: progression-free survival; OS: overall survival; m: months; y, year; CRT: concurrent chemoradiation; CT: chemotherapy; P: Pembrolizumab; A: Atezolizumab; N: nivolumab; NR: not reached; mOS: median overall survival.

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
