# Peer review of "Immunotherapy Moves to the Early-Stage Setting in Non-Small Cell Lung Cancer: Emerging Evidence and the Role of Biomarkers"

_cancers, 2020, doi:10.3390/cancers12113459_

Round 1

Reviewer 1 Report

This review article focusing the immunotherapy in the early-stage of non-small cell lung cancer is well-written and interesting. Several revisions may be helpful for the readers.

1) Title: The full-spelling is preferable for NSCLC.

2) The brief description or the table regarding the characteristics of immune checkpoint inhibitors discussed here may be helpful for the readers.

3) Some errors that should be corrected:

Line 230: “due to due to” should be “due to”.

Line 325: “circulatin” should be “circulating”.

Lines 463 and 481: “interferon 1” should be “type I interferon”.

Author Response

Thank you for your thorough revision of our manuscript, we really appreciate your comments that will sure help to improve it.

We changed the title as you suggested: we finally used the full-spelling of Non-small cell lung cancer instead of NSCLC.

We added a table with different type of ICIs discussed in the manuscript in the introduction.

We corrected pointed spelling errors.

We also added some explanations and changes suggested by the other reviewer:

-Regarding the differences between early-stage and advanced NSCLC, we added a paragraph (lines 81-85) explaining them.

-Both early-stage and advanced NSCLC could respond to immunotherapy in the same way, we added another paragraph explaining it that make patients at different stages respond differently to immunotherapy between lines 92 and 96.

-We finally removed figure 1 and 2 as they were not intuitive and information is well explained in the text.

-We added a paragraph with a brief explanation of developed biomarkers in advanced NSCLC and that same biomarkers are being explored in early-stage NSCLC, but in this later context results are only preliminary (lines 426-435).

Thank you very much for your support.

Reviewer 2 Report

The authors describe recent studies and clinical trials on the implementation of immunotherapy in patients with early-stage NSCLC. The paper includes many clinical trials. However, I have a few comments. 1. The differences between early-stage and advanced NSCLC is missing in the article. What could be the underlying mechanisms that make patients at different stages respond differently to immunotherapy? What are the possible reasons what we have for advanced NSCLC cannot apply to early-stage NSCLC? 2. Figures 1 and 2 are not intuitive or informative enough. Please remake the figures. 3. The authors discuss the role of biomarkers in the selection of best therapeutic options for patients with early-stage NSCLC. What are the biomarkers for advanced NSCLC? and whether they are suitable for early-stage NSCLC?

Author Response

Thank you for your thorough revision of our manuscript, we really appreciate your comments that will sure help to improve it.

Regarding the differences between early-stage and advanced NSCLC, we added a paragraph (lines 81-85) explaining them.

Both early-stage and advanced NSCLC could respond to immunotherapy in the same way, we added another paragraph explaining it that make patients at different stages respond differently to immunotherapy between lines 92 and 96.

We added a table with different type of ICIs discussed in the manuscript as the other peer-reviewer suggested.

We finally removed figure 1 and 2 as they were not intuitive and information is well explained in the text.

We added a paragraph with a brief explanation of developed biomarkers in advanced NSCLC and that same biomarkers are being explored in early-stage NSCLC, but in this later context results are only preliminary (lines 426-435).

Thank you very much for your support.